# Radiological Evaluation of Stainless Steel Crowns Placed on Permanent Teeth in Patients Treated under General Anaesthesia

**DOI:** 10.3390/ijerph18052509

**Published:** 2021-03-03

**Authors:** Marie-Laure Munoz-Sanchez, Natacha Linas, Nicolas Decerle, Valérie Collado, Denise Faulks, Emmanuel Nicolas, Martine Hennequin, Pierre-Yves Cousson

**Affiliations:** 1CROC, Université Clermont Auvergne, F-63000 Clermont-Ferrand, France; m-laure.munoz-sanchez@uca.fr (M.-L.M.-S.); natacha.linas@uca.fr (N.L.); nicolas.decerle@uca.fr (N.D.); valerie.collado@uca.fr (V.C.); denise.faulks@uca.fr (D.F.); emmanuel.nicolas@uca.fr (E.N.); p-yves.cousson@uca.fr (P.-Y.C.); 2CHU Clermont-Ferrand, Service d’Odontologie, F-63003 Clermont-Ferrand, France

**Keywords:** stainless steel crown, general anaesthesia, evaluation

## Abstract

Evidence-based evaluations of dental treatment are needed to support the development of special care dentistry services. This retrospective study was designed to collect and analyse X-ray images of permanent teeth restored with stainless steel crowns (SSC) in patients treated under general anaesthesia. Between 2013 and 2019, 360 permanent molars were crowned with SSCs in 198 adult patients. One calibrated investigator used an original validated tool to evaluate four radiographic criteria for molars restored with SSCs: (i) marginal adaptation; (ii) interdental proximal contact; (iii) the presence of glass ionomer cement overflow; and (iv) the loss of alveolar bone. Overall, no defect or a minor defect was reported for the majority of SSCs for the criteria “Marginal adaptation” (62.5%, *n* = 320), “Proximal contact” (82.2%, *n* = 236) and “Cement overflow” (95.8%, *n =* 337). Alveolar bone resorption was reported in 8.3% of cases, *n* = 14, after a mean period of 8.9 ± 14.3 months. It was shown that the restoration of permanent teeth using SSCs placed under general anaesthesia presents a low risk of periodontal morbidity in the medium term when assessed radiographically.

## 1. Introduction

Within dental public health, it is well recognised that persons from marginalised groups and/or disabled persons have difficulty accessing dental services for environmental, social and medical reasons [1,2,3]. It is also accepted that dental treatment sometimes has to be provided in what are qualified as “special circumstances” [4]—for example, patients requiring general anaesthesia or sedation services due to a lack of cooperation; patients requiring treatment during pandemic periods or humanitarian crises; patients that cannot or are unlikely to attend subsequent appointments due to acute medical or mental health issues; homeless patients, travellers, refugees, etc. These special circumstances have in common the need for effective treatment in a single short session. In the past, the response to dental need in special circumstances was predominantly extraction or abstention [5,6,7,8]. This response is clearly inequitable, particularly when considering that these same populations are the least likely to benefit from prosthetic rehabilitation following tooth loss [9].

Special care dentistry (SCD) is concerned with the oral health of such populations and, with the development of SCD specialties around the world [10,11,12,13], it is increasingly urgent to define effective treatment protocols for special circumstances [4,14]. Endodontic treatments, whether they be root canal treatment or full pulpotomy, have been evaluated when performed under general anaesthesia [4,15,16], but few studies have reported on restorative procedures [17,18]. Preformed stainless steel crown (SSC) restorations represent a viable treatment choice for severely carious or fractured posterior permanent teeth [19]. The use of SSCs is well described for deciduous molars with deep or extensive carious lesions [20,21,22]. It has been shown that the risk of failure or long-term pain in deciduous molars treated by pulpotomy or pulpectomy and restored with SSCs is reduced compared to direct composite restorations. Composite restorations require perfect moisture control to achieve adhesion and are time-consuming. Both these conditions may be difficult to manage in special circumstances. Few studies have explored the outcome of SSC restorations in adults [19,23]. It was recently reported that the survival of posterior permanent teeth restored with SSCs is 79.2% after 10 years. However, serious concerns are raised regarding the ability of preformed SSCs to satisfy the same goals as those required from a conventional crown. SSCs are prefabricated in a stainless nickel-chromium alloy. They are selected for use according to their adaptation to the tooth rather than being custom-made from a dental impression. One of the difficulties in placing SSCs lies in ensuring cervical adjustment whilst restoring interdental contact points. Indeed, coronal preparation for SSC placement is aggressive and marginal adaptation may be approximate. The authors are unaware of any academic guidelines regarding coronal preparation for SSCs in permanent molars. It is pertinent to consider the benefit/risk ratio for the placement of SSCs, particularly when placed in special circumstances. It is also important that the potential longer term periodontal morbidity is investigated and reported.

This study aims to evaluate the radiological outcome of SSCs placed on permanent molars for patients treated under general anaesthesia (GA).

## 2. Materials and Methods

### 2.1. Study Design

This retrospective study was designed to collect and analyse X-ray images of permanent teeth restored with SSC in patients treated under GA. Study ethics approval was obtained on the 5th of January 2021 (CECIC Rhône-Alpes-Auvergne, Grenoble, IRB 5921) and was performed in accordance with the ethical standards as laid down in the 1964 Declaration of Helsinki and its later amendments.

### 2.2. Study Population

The study population consisted of all outpatients from the special care unit of the Dental University Hospital of Clermont-Ferrand, France, treated under GA between January 2013 and April 2019. Patients treated under GA included patients presenting anxiety disorders and high levels of carious disease, medically compromised patients, patients with physical and/or intellectual disability and patients presenting with a severe gag reflex. Any patient ≥ 18 years old, for whom at least one permanent molar was restored with an SSC, and for whom at least one radiograph was available, was included in the study. A patient’s age and gender, date of SSC restoration under GA, and dates for consecutive X-ray images during the post-operative follow-up period were recorded from patient files. The follow-up period varied among patients depending on their ability to access dental care. Some patients had regular check-ups, while others were only seen when they had infectious or traumatic problems. Moreover, depending on the patient’s ability to cope with radiographic examination, post-operative images were either retro-alveolar or panoramic radiographs. Patients were examined conventionally or under relative analgesia. For patients who could not cooperate under these conditions, dental examination was delayed and completed at a later treatment session under midazolam sedation or general anaesthesia. Neither midazolam sedation nor general anaesthesia was indicated solely for the radiographic examination goals of this study.

### 2.3. Coronal Preparation for SSC Restoration

Operating protocols for conservative and endodontic treatment under GA have previously been reported [24]. SSCs for permanent teeth are available in six sizes based on the anatomy of the upper or lower first molar (3M™ Stainless Steel crowns for permanent molars). The procedure for SSC placement is illustrated in Figure 1. Following endodontic treatment (either pulpotomy or root canal treatment), the walls of the pulp chamber are cleaned (Figure 1A) and the chamber is filled with low viscosity glass ionomer cement (GIC) (Figure 1B). Once the GIC is set, the rubber dam (RD) is removed. The crown is then reduced occlusally (Figure 1C) and prepared by complete peripheral stripping to the gingival or sub-gingival level (Figure 1D). The width between the two adjacent teeth determines the size of SSC selected. The coronal preparation is then adjusted to the internal diameter of the SSC as necessary. The criteria for adjustment are (i) the cervical limits of the SSC cover the cervical limits of the prepared tooth; (ii) the marginal crest height of the SSC matches that of the adjacent teeth; (iii) proximal contacts with the adjacent teeth are restored; and (iv) occlusion is checked using 200 µm articulating paper with the jaw moved into maximal tooth contact by the operator. The SSC is filled with high or low viscosity glass ionomer cement (respectively Fuiji IX and Fuji II) and placed on the prepared tooth (Figure 1E).

### 2.4. Study Criteria

#### 2.4.1. Criteria for Radiographic Assessment

Following discussion based on the literature [19,20,25,26,27,28,29], four teachers in restorative dentistry and endodontics devised four criteria for the analysis of X-ray images of SSC restorations: (i) marginal adaptation (6 categories); (ii) proximal contact with adjacent tooth (5 categories); (iii) presence of glass ionomer cement (GIC) overflow (4 categories); and (iv) resorption of interdental alveolar bone (5 categories). Descriptive criteria for scoring each category and data interpretation are presented in Table 1. A guide was created to illustrate each category using X-ray images of SSCs (Figure 2).

The criterion “alveolar bone resorption” was evaluated by the comparison of the position of the alveolar crest between the radiograph taken at T1 during the GA session (either the preoperative X-ray, or by default, the immediate postoperative X-ray) and the most recent follow-up X-ray image (T2). The other three criteria, “marginal adaptation”, “proximal contact”, and “presence of GIC overflow”, were assessed on the most recent available X-ray collected during the routine follow-up examination (T2). Each criterion was scored twice, once for the mesial side and once for the distal side of the restored tooth.

#### 2.4.2. Calibration Tool

A calibration tool of 60 slides was created with PowerPoint software (Microsoft Corporation, Redmond, Washington, DC, USA). For each category of each of the four study criteria, three slides were created. Each slide consisted of two X-ray images: one radiograph taken at T1 during the GA session (either the preoperative X-ray, or by default, the immediate postoperative X-ray), and one radiograph taken at T2 during the most recent check-up appointment. X-ray images were resized to the same format. When multiple teeth were present on the X-ray, the tooth to be scored was indicated by an arrow and M was overprinted on the mesial face.

#### 2.4.3. Validity of the Calibration Tool

Reliability and stability of the study criteria evaluation were tested by three independent investigators, who assessed the set of 60 slides twice with a two-week interval. Intra-class coefficient for intra-rater and inter-rater evaluations are presented in Table 2 and Table 3, respectively. High ICC values demonstrated that inter and intra-investigators’ evaluations were stable and reliable.

### 2.5. Data Collection

#### 2.5.1. Radiographs of SSCs Performed under GA

Patients ≥ 18 years old, treated under GA, during which one or more SSCs were placed on permanent teeth, were identified using the hospital database. The radiographs corresponding to the SSCs of the identified patients were downloaded and anonymised. Each slide comprised two X-ray images collected from one patient record: one radiograph taken at T1 during the GA session (based on retro-alveolar or panoramic radiograph, either the preoperative X-ray, or by default, the immediate postoperative X-ray), and one radiograph taken at T2 during the latest check-up appointment (retro-alveolar or panoramic radiograph). A PowerPoint document was created in the same format as the calibration tool.

#### 2.5.2. Evaluation of Radiological Criteria

One calibrated investigator, who had participated in the validation process of the calibration tool (investigator #3), scored all the radiological criteria for the pool of X-ray images using the study criteria. The time period between the initial and the second radiograph was noted for the criterion “Alveolar bone resorption”. For statistical reasons, the follow-up times were grouped into two re-evaluation periods for ease of interpretation: <1 year and ≥1 year. Potential correlation between study criteria was tested using the Pearson correlation coefficient (*α* = 0.05).

## 3. Results

### 3.1. Study Population

Between 2013 and 2019, 360 permanent molars were crowned with SSCs in 198 patients (36.7 ± 11.2 years, from 18 to 93 years). Of the patients, 100 were men (37.7 ± 10.7 years, from 19 to 67 years) and 98 were women (35.8 ± 11.7 years, from 18 to 93 years). The reasons for being referred for dental care under GA were dental anxiety or phobia for 156 patients (78.8%), medical systemic conditions for 8 patients (4%), dementia for 1 patient (0.5%), and intellectual or physical disabilities for 33 patients (16.7%). The 360 SSCs were re-evaluated on average at 12.5 ±17.8 months (from 0.5 to 79 months). 

The flowchart of inclusions is presented in Figure 3.

### 3.2. Descriptive Results

Overall, no defect or only a minor defect was recorded for the majority of SSCs for the criteria “Marginal adaptation” (62.5%, *n* = 320), “Proximal contact” (82.2%, *n* = 236), and “Cement overflow” (95.8%, *n* = 337) (Table 4). Acceptable marginal adaptation was less often achieved than acceptable proximal contact or the absence of cement overflow.

The criterion “Alveolar bone resorption” could not be scored for 144 SSCs (40.0%) for at least one of the proximal faces. Of the 216 teeth evaluated radiologically, 194 (89.8%) had no periodontal defect (Table 5) and only 14 showed major resorption (8.3%).

Correlation was found between alveolar bone resorption and both mesial proximal contact (*r* = 0.46, *p* < 0.001) and distal proximal contact (*r* = 0.19, *p* < 0.01). There was no correlation between alveolar bone resorption and either cement overflow or marginal adaptation.

## 4. Discussion

This study reports that the restoration of permanent teeth using stainless steel crowns placed under general anaesthesia presents a low risk of periodontal morbidity, at least in the medium term. This finding is initially surprising and should be interpreted with caution, as the evaluation was based only on radiological criteria. A long-term investigation into the periodontal impact of SSCs based on clinical criteria is needed to complete the present results. Retention of dental plaque and calculus, gingival bleeding, and periodontal pocketing characterise the periodontal morbidity of SSCs clinically. However, long-term periodontal evaluation is likely to be difficult in the target population treated under GA, complicating the feasibility of such a study. Uncooperative behaviour and failure to return for follow-up could reduce the quantity and quality of data collected. Moreover, a lack of oral hygiene and high levels of medical comorbidity and medication could act as confounding factors for periodontal morbidity. Another limit of the current study is that occlusal adaptation could not be evaluated, as solely radiological criteria were used. SSCs are chosen as the restoration of choice for placement in special circumstances by our team because it is assumed that they restore dental function, i.e., that they maintain chewing function and provide a stable occlusion for swallowing [18,30]. A previous cohort study demonstrated that a comprehensive restorative approach, including SSC placement for permanent molars, helped to increase chewing activity and oral health-related quality of life following rehabilitation [18]. The current results are also in accordance with a recent, large-scale study, which concluded that SSCs are a durable treatment option for the restoration of the severely carious posterior permanent dentition, in the context of GA [19]. The original scale used in the current study for the radiological evaluation of SSCs could be used in future prospective, longitudinal investigations.

The restoration of severely damaged posterior teeth remains a key area of interest, particularly in the treatment of endodontically treated teeth. The most common restorations are full or partial crowns, with or without a cast or prefabricated post and core. Two reviews have reported that there is insufficient reliable evidence to determine which treatment is most effective [31,32]. SSCs were not considered in these reviews and uncertainty about the comparative clinical performance of SSCs or conventional crowns and fillings is related to differences in treatment concepts. The objectives of a conventional crown are to retain a functional tooth on the arch, to maintain periodontal health, and to maintain or restore proximal dental contacts. A conventional crown should protect the endodontic treatment of a restored tooth from bacterial recontamination or protect a vital tooth from carious disease and bacterial contamination. Finally, a conventional crown should restore optimal occlusal function and may have aesthetic properties. With conventional crowns, the concept is to prepare the tooth, to take an impression, and to create a prosthetic crown which is perfectly adapted to the coronal preparation. In these conditions, major defects are rare. With preformed SSCs, the concept is reversed. The tooth is adapted to the SSC, such that the SSC has the best possible proximal and occlusal contact. This exercise is difficult and requires experience. In this study, a higher frequency of major defects was reported first for the criterion “marginal adaption”, and second for “proximal contact”. Although not analysed in the data given here, it is harder to place SSCs on some teeth than others. For instance, in patients with long term loss of tooth substance, teeth may move mesially or vertically, closing the interdental space and altering the occlusion. In this case, it may be necessary to adjust the occlusion by reducing enamel thickness of the adjacent or opposing teeth, or to correct the profile of the SSC with the risk of crown perforation. In the presence of sub-gingival limits, it can be difficult to achieve continuous cervical coverage as SSCs are pre-sized in height. Seeking and applying solutions in all of these situations is time consuming and may compromise the success of SSC placement.

This study suggests that SSCs are indicated to maintain teeth on the arch and avoid extraction when conventional fixed crowns are not possible under special circumstances. In these situations, conventional indirect restorations cannot be provided because this type of restoration involves multiple steps and the intervention of a dental technician. In special circumstances, the risk–benefit ratio is not in favour of multiplying dental sessions. In addition, impression-taking is complicated under GA, particularly if orotracheal intubation is used. Most patients treated in special circumstances have difficulties accessing dental care, present high levels of dental need, high levels of decayed, missing or filled teeth (DMFT) and severely damaged teeth [18,33,34,35]. It is known that survival rates for teeth with carious cavities on more than three surfaces are higher when restored with full or partial crowns rather than with composite restorations [36]. This supports placement of SSCs rather than composite or glass ionomer cement restorations for patients treated in special circumstances. In addition, sealing SSCs with GIC may compensate for the possible lack of SSC adaption, as it has good filling and sealing properties. GIC has been shown to be better at preventing microleakage and has better retentive strength than either zinc phosphate or zinc polycarboxylate cement [37,38,39].

As the concepts and technology used in dentistry evolve, it may become possible to provide custom-made crowns even in special circumstances. Digital impressions, or computer aided design and manufacturing (CAD/CAM), associated with modern bonding techniques, could be used chairside to provide fully functional restorations that fit perfectly, all in one session [40]. For the moment, this opportunity is limited by the availability of the necessary equipment and the investment cost, but also by the limitations of bonding techniques. Bonding currently requires supra-gingival cervical margins onto enamel, and sufficient residual tooth tissue for strength. In the meantime, it is possible that SSC restoration is the next best solution to avoid extraction in permanent posterior teeth.

## 5. Conclusions

For some patients, environmental or individual conditions mean that dental treatment has to be performed in special circumstances. Although SSC placement is a well-recognised technique in the primary dentition, little information is available regarding SSC use in permanent teeth. It was therefore important to study the risks and benefits of this procedure in special circumstances. The results show that the restoration of permanent teeth using stainless steel crowns placed under general anaesthesia presents a low risk of periodontal morbidity, when assessed radiographically. This supports the placement of SSC restorations over extraction for severely damaged teeth when treatment is required in one single and short session, such as under GA, or during pandemic periods or humanitarian crises. Further studies based on clinical criteria should be undertaken to complete the current results.

## Figures and Tables

**Figure 1 ijerph-18-02509-f001:**
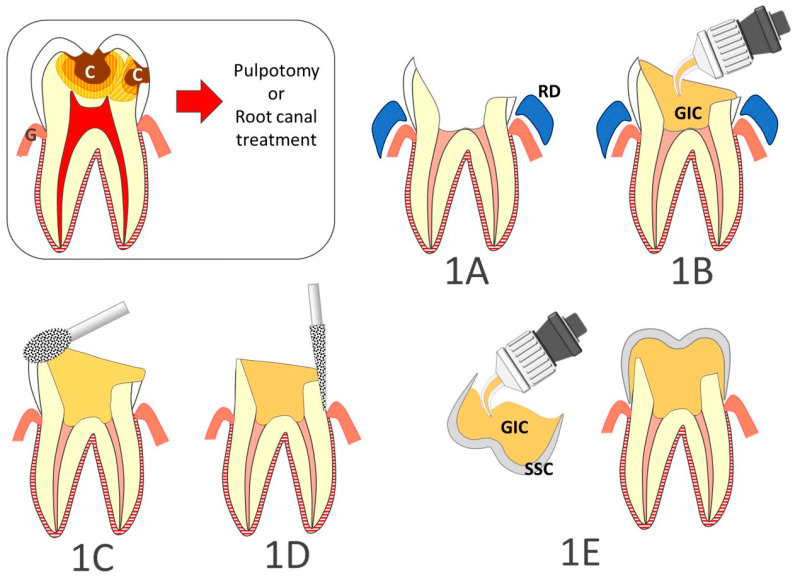
Step by step restoration of posterior teeth with large carious lesions (C) using stainless steel crowns. (**1A**): the walls of the pulp chamber are cleaned; (**1B**): the chamber is filled with low viscosity glass ionomer cement (GIC); (**1C**): the rubber dam (RD) is removed, and the crown is reduced occlusally; (**1D**): the dental crown is prepared by complete peripheral stripping to the gingival or sub-gingival level; (**1E**): The stainless stell crown (SSC) is filled with glass ionomer cement and placed on the prepared tooth. (G, Gingiva).

**Figure 2 ijerph-18-02509-f002:**
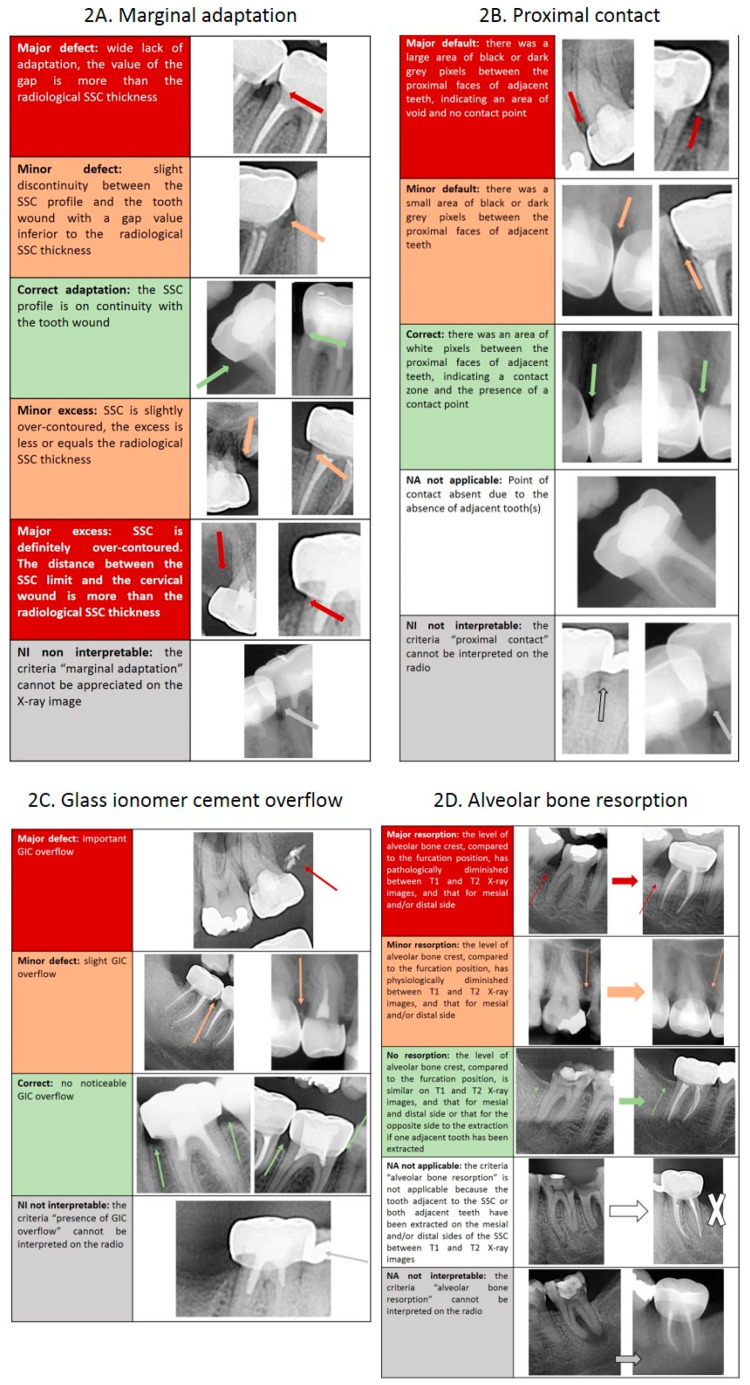
Scoring guide for calibration ((**2A**): marginal adaptation; (**2B**): proximal contact; (**2C**): presence of cement overflow; (**2D**): alveolar bone resorption).

**Figure 3 ijerph-18-02509-f003:**
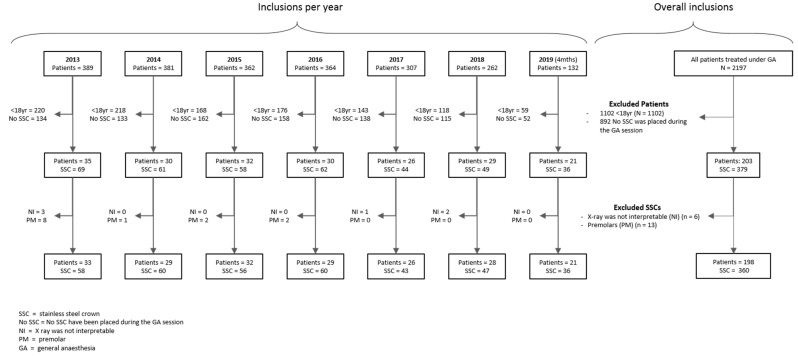
Flowchart for inclusions.

**Table 1 ijerph-18-02509-t001:** Description of the categories of the four study criteria and their data interpretation.

Criteria	Descriptive Categories	Data Interpretation
Marginal adaptation scored on T2 X-ray image	The SSC profile is well adapted to the contour of the crown preparation.	No defect or overhang
	The SSC does not reach the contour of the crown preparation, but the defect is less than the radiological thickness of the SSC.	Minor defect or overhang
	The SSC overhangs the contour of the crown preparation, but the overhang is less than the radiological thickness of the SSC.	
	The SSC does not reach the contour of the crown preparation, and the defect is greater than the radiological thickness of the SSC.	Major defect or overhang
	The SSC overhangs the contour of the crown preparation, and the overhang is greater than the radiological thickness of the SSC.	
	The criteria “marginal adaptation” cannot be evaluated on the X-ray image.	Not interpretable
Proximal contact scored on T2 X-ray image	The image shows an area of white pixels between the proximal faces of adjacent teeth, indicating a contact zone and the presence of a contact point. Clinically, there would be an appropriate contact point.	No defect
	The image shows a small area of black or dark grey pixels between the proximal faces of adjacent teeth, indicating a slight gap between adjacent teeth. Clinically, there would be a weak contact point.	Minor defect
	The image shows a large area of black or dark grey pixels between the proximal faces of adjacent teeth, indicating a wide gap and the absence of a contact point. Clinically, there would be no contact point.	Major defect
	Contact point absent due to absence of adjacent tooth or teeth.	Not applicable
	The criteria “proximal contact” cannot be evaluated on the X-ray image.	Not interpretable
Glass Ionomer Cement (GIC) overflow scored on T2 X-ray image	No noticeable GIC overflow.	No overflow
	Slight GIC overflow.	Minor overflow
	Large GIC overflow.	Major overflow
	The criteria “presence of GIC overflow” cannot be interpreted on the X-ray image.	Not interpretable
Alveolar bone resorption scored by comparing T1 and T2 X-ray images	The level of alveolar bone crest, compared to the furcation, is similar between T1 and T2, for both mesial and distal sides (or for the opposite side to the extraction if an adjacent tooth has been extracted).	No or minor resorption
	The level of alveolar bone crest, compared to the furcation, is slightly but not significantly reduced between T1 and T2, for either the mesial and/or the distal side (physiological resorption).	
	The level of alveolar bone crest, compared to the furcation, is significantly reduced between T1 and T2, for either the mesial and/or the distal side (pathological resorption).	Major resorption
	The criteria “alveolar bone resorption” is not applicable because the adjacent tooth or teeth have been extracted between T1 and T2.	Not applicable
	The criteria “alveolar bone resorption” cannot be evaluated on the X-ray image.	Not interpretable

**Table 2 ijerph-18-02509-t002:** Intra-class correlation coefficient (ICC) for intra-rater validity for each study criterion (n: number of cases).

	Mesial Maginal Adaptation	Distal Marginal Adaptation	Mesial Proximal Contact	Distal Proximal Contact	Mesial GIC Overflow	Distal GIC Overflow	Mesial Alveolar Bone Resorption	Distal Alveolar Bone Resorption
Investigator #1	0.973 *n* = 56	0.968 *n* = 54	0.955 *n* = 54	0.999 *n* = 55	0.936 *n* = 55	1 *n* = 53	0.914 *n* = 57	0.999 *n* = 55
Investigator #2	0.766 *n* = 59	0.781 *n* = 57	0.999 *n* = 52	0.999 *n* = 56	0.882 *n* = 60	0.848 *n* = 60	0.930 *n* = 46	0.946 *n* = 46
Investigator #3	0.906 *n* =52	0.781 *n* = 51	0.998 *n* = 49	0.999 *n* = 54	0.936 *n* = 54	0.92 *n* = 53	0.995 *n* = 49	0.992 *n* = 52

**Table 3 ijerph-18-02509-t003:** Intra-class correlation coefficient (ICC) for inter-rater assessment for each study criterion (*n*: number of cases).

	Mesial Marginal Adaptation	Distal Marginal Adaptation	Mesial Proximal Contact	Distal Proximal Contact	Mesial GIC Overflow	Distal GIC Overflow	Mesial Alveolar Bone Resorption	Distal Alveolar Bone Resorption
Investigator #1 vs. Investigator #2	0.818 *n* = 56	0.762 *n* = 54	0.952 *n* = 52	0.935 *n* = 54	0.760 *n* = 55	0.920 *n* = 53	0.827 *n* = 49	0.984 *n* = 50
Investigator#1 vs. Investigator #3	0.671 *n* = 53	0.603 *n* = 51	0.972 *n* = 51	0.934 *n* = 54	0.807 *n* = 54	0.824 *n* = 52	0.831 *n* = 52	0.91 *n* = 52
Investigator #2 vs. Investigator #3	0.624 *n* = 54	0.619 *n* = 53	0.995 *n* = 51	0.998 *n* = 56	0.826 *n* = 54	0.757 *n* = 56	0.984 *n* = 47	0.903 *n* = 48

**Table 4 ijerph-18-02509-t004:** Evaluation of SSCs for “Marginal Adaptation”, “Proximal contact”, and “Cement overflow”.

	Marginal Adaptation (*n* = 320 SSCs)	Proximal Contact (*n* = 236 SSCs)			Cement Overflow (*n* = 337 SSCs)
		Mesial and Distal	Mesial or Distal If Absence of Adjacent Tooth	Sub-Total	
No defect	94 (29.4%)	46 (19.5%)	97 (41.1%)	143 (60.6%)	270 (80.1%)
Minor defect	106 (33.1%)	22 (9.3%)	29 (12.3%)	51 (21.6%)	53 (15.7%)
Major defect	120 (37.5%)	11 (4.7%)	31 (13.1%)	42 (17.8%)	14 (4.2%)

**Table 5 ijerph-18-02509-t005:** Evaluation of SSCs for “Alveolar bone resorption” (*n* = 216).

	<12 Months (2 ± 2.4 Months)	≥12 Months (31.8 ± 13.7 Months)	Total (8.9 ± 14.3 Months; Max: 0.5 Months Min: 66 Months)
No resorption or minor resorption for both mesial and distal faces	99 (76.2%)	31 (23.8%)	130 (100%)
No resorption or minor resorption for one face and extraction of the adjacent tooth for the other face	52 (81.3%)	12 (18.7%)	64 (100%)
Major resorption	8 (57.1%)	6 (42.9%)	14 (100%)
Not applicable because of extraction of both adjacent teeth	7 (87.5%)	1 (12.5%)	8 (100%)
Total	166 (76.9%)	50 (23.1%)	216 (100%)

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
