# Peer review of "Radiological Evaluation of Stainless Steel Crowns Placed on Permanent Teeth in Patients Treated under General Anaesthesia"

_ijerph, 2021, doi:10.3390/ijerph18052509_

Round 1

Reviewer 1 Report

This study evaluated the stainless steel crowns placed on permanent teeth in patients treated under general anaesthesia, which provides interesting result and good evidence for clinical practice. The presentation of methods and results was clear. But there are several issues needed to be addressed.

  1. For the x ray, it did not indicate what type of radiograph it was, such as bitewing or pA? Also whether the authors checked the condition of the adjacent teeth, such as presence of proximal caries, root resorption which could be caused by the compromised SSC quality, and this data would make the result more comprehensive.
  2. Regarding the result analysis, was there any correlation among the study criteria? For example, did the tooth with major defect in bone resorption also had large cement overflow?
  3. Regarding the discussion, there were some repetition to the Introduction section, such as the description on "special circumstance" and the advantage of SSC placement under GA.
  4. There should be more thorough discussion on the results. For example, what is the possible reason of the major defect? Does it relate to the clinical fault, patient's age, tooth number (accessibility), periodontal condition (bone loss on radiograph), medical condition etc?
  5. According to this study, SSC can be a good option for patient in special circumstance. Is there any clinical advice from the authors on the selection criteria of tooth accepting SSC (in which condition the tooth has higher success rate after SSC placement), and what should be considered to minimize the major defect?
  6. The study ethics approval was obtained on 14 December 2020 and the manuscript was received in 2020. It seems the calibration, data collection and analysis and manuscript preparation was completed within 2 weeks, which was a very tight timeframe. Please explain or confirm the approval date.
  7. Please double check the format of reference. There are some inconsistencies, such as page number.

Author Response

  1. For the x ray, it did not indicate what type of radiograph it was, such as bitewing or pA?

Answer: we added the following paragraph to describe the type and conditions for X-ray images recording

Lines 86-93

The follow-up period varied among patients depending on their ability to access dental care. Some patients had regular check-ups, while others were only seen when they had infectious or traumatic problems. Moreover, depending on the patient’s ability to cope with radiographic examination, post-operative images were either retro-alveolar or panoramic radiographs. Patients were examined conventionally or under relative analgesia. For patients who could not cooperate under these conditions, dental examination was delayed and completed at a later treatment session under midazolam sedation or general anaesthesia. Neither midazolam sedation nor general anaesthesia was indicated solely for the radiographic examination goals of this study.

Also whether the authors checked the condition of the adjacent teeth, such as presence of proximal caries, root resorption which could be caused by the compromised SSC quality, and this data would make the result more comprehensive.

Answer: We agree, and this was a point of discussion among the authors during the development of the calibration tool. Finally, we decide to not retain the criteria “presence of a carious lesion” due to the difficulty of linking the carious process to the single factor “presence of a preformed crown”. In this population of patients with a very high caries risk, there are too many factors to be controlled before cause and effect could be inferred between the SSCs and the development of new carious lesions.

  1. Regarding the result analysis, was there any correlation among the study criteria? For example, did the tooth with major defect in bone resorption also had large cement overflow?

Answer: The possible correlations were searched for. We added the following sentence in the paragraph “statistical analysis”

Lines 186-188

Correlation was found between alveolar bone resorption and both mesial proximal contact (r=0.46, p<0.001) and distal proximal contact (r=0.19, p<0.01). There was no correlation between alveolar bone resorption and cement overflow or marginal adaptation.

  1. Regarding the discussion, there were some repetition to the Introduction section, such as the description on "special circumstance" and the advantage of SSC placement under GA.

Answer: We agree. Repetition has been deleted from the discussion section lines 198-205

  1. There should be more thorough discussion on the results. For example, what is the possible reason of the major defect? Does it relate to the clinical fault, patient's age, tooth number (accessibility), periodontal condition (bone loss on radiograph), medical condition etc?

and

  1. According to this study, SSC can be a good option for patient in special circumstance. Is there any clinical advice from the authors on the selection criteria of tooth accepting SSC (in which condition the tooth has higher success rate after SSC placement), and what should be considered to minimize the major defect?

Answer:  Discussion of these matters has been added to the manuscript. lines 212-234

The restoration of severely damaged posterior teeth remains a key area of interest, particularly in the treatment of endodontically treated teeth. The most common restorations are full or partial crowns, with or without a cast or prefabricated post and core. Two reviews have reported that there is insufficient reliable evidence to determine which treatment is most effective [31,32]. SSCs were not considered in these reviews and uncertainty about the comparative clinical performance of SSCs or conventional crowns and fillings is related to differences in treatment concepts. The objectives of a conventional crown are to retain a functional tooth on the arch, to maintain periodontal health and to maintain or restore proximal dental contacts. A conventional crown should protect the endodontic treatment of a restored tooth from bacterial recontamination or protect a vital tooth from carious disease and bacterial contamination. Finally, a conventional crown should restore optimal occlusal function and may have aesthetic properties. With conventional crowns, the concept is to prepare the tooth, to take an impression and to create a prosthetic crown which is perfectly adapted to the coronal preparation. In these conditions, major defects are rare. With preformed SSCs, the concept is reversed. The tooth is adapted to the SSC, such that the SSC has the best possible proximal and occlusal contact. This exercise is difficult and requires experience. In this study, a higher frequency of major defects was reported first for the criterion “marginal adaption”, and second for “proximal contact”. Although not analysed in the data given here, it is harder to place SSCs on some teeth than others. For instance, in patients with long term loss of tooth substance, teeth may move mesially or vertically, closing the interdental space and altering the occlusion. In this case, it may be necessary to adjust the occlusion by reducing enamel thickness of the adjacent or opposing teeth, or to correct the profile of the SSC with the risk of crown perforation. In the presence of sub-gingival limits, it can be difficult to achieve continuous cervical coverage as SSCs are pre-sized in height. Seeking and applying solutions in all of these situations is time consuming and may compromise the success of SSC placement.

  1. The study ethics approval was obtained on 14 December 2020 and the manuscript was received in 2020. It seems the calibration, data collection and analysis and manuscript preparation was completed within 2 weeks, which was a very tight timeframe. Please explain or confirm the approval date.

Answer: The study was conducted according to the ethical frame devoted to the Use of Routine Data in Health Research. For such studies, the ethical board of our establishment asks clinicians to request ethical approval once the manuscript is ready to be submitted for publication, or to a wider extent, for any dissemination of results. For the present study we received a decision from our ethical board in two steps. Firstly, we submitted the manuscript to the ethical committee on the 2nd of December 2020 and we received an answer on the 14th of December (Supplementary file 1). At this step, the ethical committee asked for the insertion of the sentence: “All patients and their carers were given information about the possible use for scientific goals of the data issued from their dental treatment” in the study design paragraph lines74-75. This insertion was made and the manuscript submitted. Secondly, during the IJERPH reviewing process, we received the final approval of the ethical committee on the 05/01/2021 (Supplementary file 2). Consequently, the date of study ethics approval given in the manuscript is now the 5th of January 2021 (lines 74-75).

  1. Please double check the format of reference. There are some inconsistencies, such as page number

Answer:  We agree. We corrected the format on the editor manager website.

Reviewer 2 Report

Authors reported a retrospective study that analyzed, in the period between 2013 and 2019, 360 permanent molars restored with stainless steel crowns in patients treated under general anesthesia. The effects were evaluated through four radiographical criteria: i) marginal adaptation; ii) interdental proximal contact; iii) presence of glass ionomer cement overflow and iv) loss of alveolar bone.  

This is an interesting study regarding dental public health in which, sometimes, dental treatments were provided in circumstances that required a single short session.  

However, there are some aspects that need to be improved:

  • Introduction
  • Line 45. Please add some references to support this sentence “Preformed stainless steel crown (SSC) restorations represent a viable treatment choice for severely carious or fractured posterior permanent teeth”.
  • Line 47. Please add some references to support this sentence “The use of SSCs is well described for deciduous molars with deep or extensive carious lesions”.
  • “It has been shown that the risk of failure or long-term pain of deciduous molars treated by pulpotomy or pulpectomy and restored with SSCs is reduced compared to direct composite restorations” please explain why the risk of failure is reduced.
  • Line 51. Please add some references to support this sentence “Few studies have explored the 51 outcome of SSC restorations in adults”.
  • Authors reported the main characteristics of conventional and stainless steel crowns (lines 57-66). However, they should explain the reasons why clinicians should choose stainless steel crowns instead conventional ones.
  • Line 68. Please rephrase this sentence “It is also important to that the potential longer term periodontal morbidity is investigated and reported”.
  • Materials and methods
  • Line 83, “Dental University Hospital of Clermont-Ferrand”, please add the state.
  • Figure 1: please add to the figure the legend “1a, 1b, 1c and 1d”, described in description.
  • Author must clarify after which timing they analysed the interdental alveolar bone resorption (T2) since the sentence “the most recent follow-up X-ray image” cannot be considered as a scientific tool (see lines 115-117). Moreover, at lines 152-153, they reported that “The follow-up times were grouped into two re-evaluation periods for ease of interpretation: < 1 year and ≥ 1 year”. They must clarify in materials and methods section after how many years maximum the X-Ray images were performed.
  • With what dental cement crowns were fixed? Author must explain.
  • Results and Discussion
  • Please correct Table 4: some words must be bold.
  • Lines 191-193, “SSCs are chosen as the restoration of choice for placement in special circumstances by our team because it is assumed that they restore dental function i.e. that they maintain chewing function and provide a stable occlusion for swallowing”, please add some references.
  • Author must add more references in order to better define the state of art and to support their study in the discussion section.

Author Response

  1. Introduction

Line 45. Please add some references to support this sentence “Preformed stainless steel crown (SSC) restorations represent a viable treatment choice for severely carious or fractured posterior permanent teeth”.

Answer:  Done

Sigal, A.V.; Sigal, M.J.; Titley, K.C.; Andrews, P.B. Stainless steel crowns as a restoration for permanent posterior teeth in people with special needs: A retrospective study. J Am Dent Assoc. 2020, 151, 136‑44.

  1. Line 47. Please add some references to support this sentence “The use of SSCs is well described for deciduous molars with deep or extensive carious lesions”.

Answer:  Done

Chen K, Lei Q, Xiong H, Chen Y, Luo W, Liang Y. A 2-year clinical evaluation of stainless steel crowns and composite resin restorations in primary molars under general anaesthesia in China's Guangdong province. Br Dent J. 2018 Jul 13;225(1):49-52. doi: 10.1038/sj.bdj.2018.519. PMID: 30002536.

Innes, N.P.T.; Ricketts, D.; Chong, L.Y.; Keightley, A.J.; Lamont, T.; Santamaria, R.M. Preformed crowns for decayed primary molar teeth. Cochrane Database Syst Rev. 2015.

Schüler IM, Hiller M, Roloff T, Kühnisch J, Heinrich-Weltzien R. Clinical success of stainless steel crowns placed under general anaesthesia in primary molars: an observational follow up study. J Dent. 2014 Nov;42(11):1396-403. doi: 10.1016/j.jdent.2014.06.009. Epub 2014 Jun 30. PMID: 24994618.

  1. “It has been shown that the risk of failure or long-term pain of deciduous molars treated by pulpotomy or pulpectomy and restored with SSCs is reduced compared to direct composite restorations” please explain why the risk of failure is reduced.

Answer: Arguments were inserted as follows, lines 51-53

Composite restorations require perfect moisture control to achieve adhesion and are time-consuming. Both these conditions may be difficult to manage in special circumstances.

  1. Line 51. Please add some references to support this sentence “Few studies have explored the outcome of SSC restorations in adults”.

Answer: Done

Zagdwon AM, Fayle SA, Pollard MA. A prospective clinical trial comparing preformed metal crowns and cast restorations for defective first permanent molars. Eur J Paediatr Dent. 2003 Sep;4(3):138-42. PMID: 14529335.

Sigal, A.V.; Sigal, M.J.; Titley, K.C.; Andrews, P.B. Stainless steel crowns as a restoration for permanent posterior teeth in people with special needs: A retrospective study. J Am Dent Assoc. 2020, 151, 136‑44.

  1. Authors reported the main characteristics of conventional and stainless steel crowns (lines 57-66). However, they should explain the reasons why clinicians should choose stainless steel crowns instead conventional ones.

Answer:  See the discussion section lines 234-251

  1. Line 68. Please rephrase this sentence “It is also important to that the potential longer term periodontal morbidity is investigated and reported”.

Answer: The superfluous ‘to’ has been removed from this sentence

  1. Materials and methods

Line 83, “Dental University Hospital of Clermont-Ferrand”, please add the state.

Answer:  Done

  1. Figure 1: please add to the figure the legend “1a, 1b, 1c and 1d”, described in description.

Answer:  Done

  1. Author must clarify after which timing they analysed the interdental alveolar bone resorption (T2) since the sentence “the most recent follow-up X-ray image” cannot be considered as a scientific tool (see lines 115-117). Moreover, at lines 152-153, they reported that “The follow-up times were grouped into two re-evaluation periods for ease of interpretation: < 1 year and ≥ 1 year”. They must clarify in materials and methods section after how many years maximum the X-Ray images were performed.

Answer:  line 169

The 360 SSCs were re-evaluated on average at 12.5 ±17.8 months (from 0.5 to 79 months).

For the criterion ‘alveolar bone resorption” we added the minimal and maximal values of the duration between radiographs in Table 5: (Max: 0.5 months, Min: 66 months)

  1. With what dental cement crowns were fixed? Author must explain.

Answer: This is now detailed in the methods and discussed as below.

Lines 251-254

In addition, sealing SSCs with GIC may compensate for the possible lack of SSC adaption, as it has good filling and sealing properties. GIC has been shown to be better at preventing micoleakage and has better retentive strength than either zinc phosphate or zinc polycarboxylate cement (Memarpour M et al 2011, Reddy et al 1998, Khinda et al 2002).

Memarpour M, Mesbahi M, Rezvani G, Rahimi M. Microleakage of adhesive and nonadhesive luting cements for stainless steel crowns. Pediatr Dent. 2011 Nov-Dec;33(7):501-4. PMID: 22353410.

Reddy R, Basappa N, Reddy VV. A comparative study of retentive strengths of zinc phosphate, polycarboxylate and glass ionomer cements with stainless steel crowns--an in vitro study. J Indian Soc Pedod Prev Dent. 1998 Mar;16(1):9-11. PMID: 11813717.

Khinda VI, Grewal N. Retentive [correction of Preventive] efficacy of glass ionomer, zinc phosphate and zinc polycarboxylate luting cements in preformed stainless steel crowns: a comparative clinical study. J Indian Soc Pedod Prev Dent. 2002 Jun;20(2):41-6. PMID: 12435014.

  1. Results and Discussion

Please correct Table 4: some words must be bold.

Answer:  Done

  1. Lines 191-193, “SSCs are chosen as the restoration of choice for placement in special circumstances by our team because it is assumed that they restore dental function i.e. that they maintain chewing function and provide a stable occlusion for swallowing”, please add some references.

Answer:  Done

Bourdiol P, Hennequin M, Peyron MA, Woda A. Masticatory Adaptation to Occlusal Changes. Front Physiol. 2020 Apr 3;11:263. doi: 10.3389/fphys.2020.00263. PMID: 32317982; PMCID: PMC7147355.

Decerle, N.; Cousson, P.Y.; Nicolas, E.; Hennequin, M. A Comprehensive Approach Limiting Extractions under General Anesthesia Could Improve Oral Health. Int J Environ Res Public Health. 2020, 17.

  1. Author must add more references in order to better define the state of art and to support their study in the discussion section.

Answer: Done in the added paragraph in the discussion section.